# A Synthetic Bio-Absorbable Membrane in Guided Bone Regeneration in Dehiscence-Type Defects: An Experimental In Vivo Investigation in Dogs

**DOI:** 10.3390/bioengineering10070841

**Published:** 2023-07-15

**Authors:** Rafael Pla, Javier Sanz-Esporrin, Fernando Noguerol, Fabio Vignoletti, Pablo Gamarra, Mariano Sanz

**Affiliations:** 1Faculty of Dentistry, University Complutense of Madrid (UCM), 28040 Madrid, Spain; 2ETEP (Etiology and Therapy of Periodontal and Peri-Implant Diseases) Research Group, Faculty of Dentistry, University Complutense of Madrid (UCM), 28040 Madrid, Spain

**Keywords:** guided bone regeneration, synthetic barrier membrane, polylactic acid and acetylbutylcitrate, dental implant, animal model

## Abstract

This study aimed to determine the performance and characteristics of a synthetic barrier membrane of polylactic acid and acetyl butyl citrate (PLAB) for the lateral bone augmentation of peri-implant dehiscence defects (mean height × depth = 3 mm × 1 mm). In eight dogs, three treatment groups were randomly allocated at each chronic peri-implant dehiscence-type defect: (i) a deproteinized bovine bone mineral covered by a synthetic barrier membrane (test group), (ii) a deproteinized bovine bone mineral covered by a natural collagen membrane (positive control), and (iii) a synthetic barrier membrane (negative control). After 4 and 12 weeks of submerged healing, dissected tissue blocks were processed for calcified and decalcified histological analysis. Histometric measurements for tissue and bone width were performed, and bone-to-implant contact and alkaline phosphatase expression where measured. After 4 and 12 weeks of healing, no statistical differences between the groups were observed for the histometric measurements. The expression of alkaline phosphatase was higher in the positive control group after 4 weeks followed by the positive and negative controls (5.25 ± 4.09, 4.46 ± 3.03, and 4.35 ± 2.28%, *p* > 0.05) and 12 weeks followed by the negative and positive controls (4.3 ± 2.14, 3.21 ± 1.53, and 2.39 ± 1.03%, *p* > 0.05). Concerning the bone-to-implant contact, after 4 weeks, the test group obtained the highest results (39.54 ± 48.7) vs. (31.24 ± 42.6) and (20.23 ± 36.1), respectively, while after 12 weeks, the positive control group obtained the highest Bone to imaplant contact (BIC) results, followed by the test and negative controls, (35.91 ± 24.9) vs. (18.41 ± 20.5) and (24.3 ± 32.1), respectively; no statistically significant differences were obtained. Within the limitations of the study, new bone formation can be achieved in guided bone regeneration procedures simultaneously with implant placement either with the use of a PLAB membrane or a native collagen membrane, although these differences were not statistically significant.

## 1. Introduction

The rehabilitation of lost dentition with dental implants has demonstrated high long-term success rates [1,2]; this success, however, is dependent on the bone availability in the alveolar ridge. A sufficient bone volume is not only needed to obtain implant stability but to enable a correct prosthetically driven tridimensional implant position, thus providing an acceptable functional and aesthetic outcome [3]. Bone availability is frequently jeopardized by the causes of tooth loss, since frequently, the presence of chronic/acute infections or trauma results in bone defects in the healed ridge. Even the uneventful healing after tooth extraction results in significant dimensional changes in the alveolar ridge that may condition an appropriate implant placement [1,4,5].

To overcome these limitations in bone availability, which is more frequently in the horizontal dimension, simultaneous implant placement with different bone regeneration interventions for lateral bone augmentation [6] has demonstrated successful outcomes [7]. Among these interventions, guided bone regeneration (GBR) combining a bone replacement graft (BRP) and a barrier membrane (BM) has not only been the most evaluated procedure but the one demonstrating higher predictability [8]. Among the different biomaterials used as BRPs, deproteinized bovine bone mineral (DBBM) has shown the highest predictability, with it being the most used BRP in GBR for lateral bone augmentation procedures [8]. Among the available BMs, bio-absorbable membranes have become the gold standard for lateral bone augmentation, with those made out of natural native porcine collagen (NPCM) being the most frequently used due to their proven properties of biocompatibility and cellular exclusion, together with easy surgical handling [9]. However, these natural collagen BMs have shown fast bio-absorbability during postoperative healing, with a reduction to half of their thickness in the first 2–4 weeks and them being completely resorbed between 4 to 12 weeks [10,11,12]. Furthermore, these forms of NPCM lack structural stiffness, which makes them collapse under light mechanical forces [13].

To overcome limitations from NPCM, cross-linked collagen membranes (CM) has been proposed as an alternative. Cross-linked CM enhances stability and resistance to degradation, ensuring long-term structural support during the healing process [10,11,14,15]. Nevertheless, membrane cross-linking increases the risk of post-operative complications like foreign body reactions, slower vascularization, inflammation, and diminished tissue integration [16] and often leads to reduced bone regeneration [17].

To improve these properties, there has been a search for synthetic bio-absorbable barrier membranes made of different polymers, such as poly(lactic acid) (PLA), poly(glycolic acid) (PGA), poly(epsilon-caprolactone) (PCL), poly(hydroxyl Valerie acid), poly (hydroxyl butyric acid), and copolymers [18].

As part of this initiative, a synthetic barrier membrane was manufactured combining polylactic acid and acetyl butyl citrate (PLAB) with the goal of providing controlled bio-absorbability and greater stiffness [19]. Although the use of a barrier membrane with this composition has been widely used and documented for the periodontal regeneration of intrabony defects and furcations [20,21], there is no information on how a BM with the same composition but designed for bone regeneration would behave, especially in lateral augmentation procedures simultaneous with implant placement [22,23].

It was therefore the aim of this investigation to evaluate the histological behavior and healing under these synthetic barrier membranes when used in experimental dehiscence-type bone defects. Our primary outcome variables were histometric variation in bone and tissue and our secondary outcomes were bone-to-implant contact and the expression of alkaline phosphatase. With this aim, we designed a prospective randomized experimental in vivo investigation comparing a test membrane (PLAB) with a control (NPCM) membrane to reconstruct chronified dehiscence-type bone defects.

## 2. Materials and Methods

The present work is a complementary analysis to the previously published Micro CT and profilometric data analysis [24]. The present report provides the histological results of the same experimental study.

The present study reports the results of an experimental in vivo investigation comparing a test membrane (PLAB) with a control (NPCM) membrane in the early (4 weeks) and delayed (12 weeks) regeneration of peri-implant dehiscence defects, with the primary outcome being the newly formed bone thickness in the exposed buccal implant surface.

This experimental in vivo investigation was approved by the Regional Ethics Committee for Animal Research (EXP-20170327) and fulfilled the ARRIVE guidelines for animal experimentation [25]. All surgical procedures were conducted at the Minimally Invasive Surgery Centre Jesús Usón in Caceres, Spain, an internationally accredited center for experimental research where specialist veterinary doctors monitored and cared for the animal’s well-being during the entire investigation.

### 2.1. Study Sample

Eight female beagles (12–24 months in age), weighing between 10 and 15 kg, were selected and monitored 4 weeks before the study to assess their general health status. Each experimental animal was identified by a subcutaneous chip, maintained in an individual kennel under a light/darkness cycle of 12:12 and a controlled temperature of 21–22 °C, and monitored daily by an experienced veterinarian. They were fed with hard pellets specifically manufactured for this species and with free access to water.

### 2.2. Surgical Interventions

Three different GBR interventions were tested at both, early (4 weeks) and late (12 weeks), healing periods [9,24].

The test group using deproteinized bovine bone mineral (DBBM) (BioOss^®^, Geistlich Pharma, Wolhusen, Switzerland) combined with a synthetic polylactic membrane (PLAB) (GUIDOR^®^, Sunstar, Schlieren, Switzerland)The positive control group using DBBM (BioOss^®^, Geistlich Pharma, Wolhusen, Switzerland) combined with a natural porcine collagen membrane (NPCM) (BioGide^®^, Geistlich Pharma, Wolhusen, Switzerland)The negative control group using only the synthetic polylactic membrane (PLAB) (GUIDOR^®^, Sunstar, Schlieren, Switzerland)

In the three similar peri-implant dehiscence-type defects created in each hemimandible, the three treatment groups were tested for either early or delayed healing (4 and 12 weeks, respectively). A block randomized allocation using a computer-generated list assigned the treatment groups depending on the defect position (mesial, medial, or distal) and the hemimandible side (left or right) (Figure 1).

In all of the surgical interventions, the animals were sedated with propofol (2 mg/kg/i.v., Propovet, Abbott Laboratories, Kent, UK), and then, general anesthesia was administrated with 2.5–4% of isoflurane (Isoba-vet, Schering-Plough, Madrid, Spain). Local anesthesia with vasoconstriction (lidocaine 2% with epinephrine 1:100,000) (2% Xylocaine Dental, Dentsply, York, PA, USA) was then infiltrated in the surgical area during the surgery.

The first surgical Intervention consisted of the creation of experimental defects for the late-healing group. The mucoperiostal flaps were elevated, and the mesial roots of the first molar and fourth premolar and the distal roots of the second and third premolar were carefully extracted. Then, the remaining roots were pulp capped with calcium hydroxide (Dycal, Dentsply, York, PA, USA), and the buccal bone plate of the fresh extraction sockets was removed, thus creating three two-wall bone defects per hemimandible (10 mm × 10 mm × 5 mm) to experimentally induce a narrow ridge. The flaps were then sutured with bio-absorbable sutures to ensure primary soft tissue healing (Figure 2).

After 8 weeks of undisturbed healing, the second surgery in this hemimandible was aimed at the simultaneous implant placement and bone regenerative intervention for the delayed healing group (12 weeks), while in the contralateral hemimandible, identical defects were created for the early healing group (4 weeks).

In these chronified bone defects (two-wall narrow ridges), two dental implants of 2.5 mm in diameter and 7 or 9 mm in length (Dentium NR^®^, Suwon, Republic of Korea) were placed in each defect, resulting in a buccal bone dehiscence around each implant. After measuring the height and width of the dehiscence with a periodontal probe, bone cortical perforations with a small round burr under profuse saline irrigation were made around the dehiscence defects. Then, the experimental treatment groups were randomly allocated, and the corresponding bone replacement grafts and barrier membranes were customized and adapted to properly fill the defects. The membranes were fixed with two metal tacks (Dentium, Seoul, Republic of Korea) to avoid displacement during healing; then, the flaps were closed with bio-absorbable sutures, after resealing the periosteum to assure a passive primary closure (Figure 3).

After 8 weeks, the same GBR surgical intervention was only carried out on the contralateral side for the 4-week healing group. The contralateral hemimandible remained in the healing process.

Four weeks after this third surgery, the experimental animals were euthanized with a lethal dose of sodium pentothal. Then, the mandibles were extracted, and the tissue blocks were dissected and placed in 10% buffered formalin for histological analysis.

### 2.3. Histological Analysis

Each fixed tissue block contained one experimental defect that included two implants. These blocks were sectioned into two halves, each containing one implant and one dehiscence defect, and were processed for either decalcified histology or undecalcified ground sectioning. The undecalcified ground sections were processed using the methodology described by Donath and Breuner [26]. In brief, the fixed specimens were first dehydrated in a graded series of ethanol solutions and embedded in a light-curing resin (Technovit 7200 VLC; Heraeus-Kulzer GMBH, Werheim, Germany). From each specimen, one buccolingual section representing the central implant area was micro-ground and polished using a band saw (Exakt^®^ Apparatebau, Norderstedt, Germany) and 1200 and 4000 grit silicon carbide paper (Struers, Copenhagen, Denmark) until achieving a thickness of approximately 30 μm. The obtained sections were then stained using Mason Goldner’s trichrome staining. These histological sections were analyzed using a Leica DMRBE microscope equipped with a Leitz DMRD micro-photographic unit (Leica Microsystems GmbH, Wetzlar, Germany) connected to a digital camera and a computer. High-resolution images were acquired and measured using a dedicated image analysis software (Image Pro Premiere 9.1; Media Cybernetic Inc., Rockville, Maryland). The histometric measurements were performed at 10× magnification using a calibrated digital tool (“smart segmentation tool” (SST)) incorporated into the image analysis software (Image pro premier 9.1). 

The following horizontal linear measurements were obtained at the buccal aspect of the implant, perpendicular to its long axis (Figure 4):NBT: Newly formed bone thickness at 1, 2, and 3 mm apical to the implant shoulder, described as the distance from the implant surface and the most buccal bone tissue.AGT: Augmented tissue thickness at 1, 2, and 3 mm apical to the implant shoulder (barrier space maintenance capacity), described as the distance from the implant surface to the inner part of the identified barrier membrane.

The percentage of bone-to-implant contact (BIC) was also calculated along the buccal aspect of the implants.

### 2.4. Decalcified Histology

For the immunohistochemical preparation, we processed the samples according to the fracture technique [27]. In brief, the tissue blocks were decalcified with EDTA (12.5%, pH = 7), which was replaced every week for 8 months. Once decalcified, the tissue blocks were split into vestibular and lingual sides, with the dental implant being removed very carefully. The resultant vestibular pieces were decalcified for two extra months with a different decalcifier (*Osteosoft* ^®^, *Merck*; Branchburg, New Jersey, USA) and then embedded in paraffin. The resulting blocks were cut into 7 µm-thick sections and stained with hematoxylin and eosin (*PanReac AppliChem*; Darmstadt, Germany). The remaining slides were used for immunohistochemical staining.

Immunohistochemical staining was carried out with the “*Master Polymer Plus Detection System (Peroxidase)*” kit (*Master Diagnóstica*; Granada, Spain), which uses rabbit monoclonal and polyclonal primary antibodies and DAB (3,3′-diaminobenzidine) as visualization solution. Specific antibodies for alkaline phosphatase (1:200 *Santa Cruz Biotechnology Inc*.; Santa Cruz, CA, USA) were used to measure bone metabolism.

### 2.5. Immunohistochemical Analysis

All slides were observed using a light microscope (*Leica Geosystems AG*; Heerbrugg, Switzerland) and photographed by ×2.5 magnification with a digital camera (*Leica Geosystems AG,* Heerbrugg, Switzerland) in a standardized light condition. A 3 mm × 3 mm square region of interest (ROI) was delimited, taking the implant shoulder as a reference [28]. The stained intensity (%) of the ALP was measured using an IHC profiler and Image J (NIH, Bethesda, Bethesda, Maryland, USA), for the quantitative analysis [29,30]. The results were divided into four categories according to the intensity of staining: high positive, positive, low positive, and negative. To reduce false positives, only high positive and positive stained values were analyzed.

### 2.6. Data Analysis

Data from continuous outcome variables were expressed as means and standard deviations. Shapiro-Wilk normality tests were used to assess the data distribution. Whenever normality could be assumed, ANOVA tests were used for the intergroup comparisons in each of the healing time point groups (4 weeks and 12 weeks). When the data did not accomplish normality, the Kruskall-Wallis test was performed to assess the intergroup comparisons. Bonferroni corrections were performed for multiple comparisons. The alpha error was set at 0.05, and all tests were performed with the statistical software package (IBM SPSS Statistics^®^ V20 JM.Domenech).

## 3. Results

The healing after the surgical interventions was uneventful without the advent of any wound dehiscence or other local adverse events. All implants and the grafted materials healed uneventfully and resulted in 48 defects being prepared for ground undecalcified sections and analyses (16 tests, 16 positives, and 16 negative control groups), with 24 for each healing period (4 and 12 weeks). Differences in the surgical handling of the tested BMs were noticeable, with the PLAB membrane being stiffer and better suited to maintain the space underneath, even without the use of a bone replacement graft (Figure 5).

### 3.1. Histological Observations

Bone regeneration up to the implant platform was never observed in any of the treatment groups. In the groups using a bone replacement graft, DBBM particles were found surrounded by new bone, although in the outer parts of the regenerated area, the granules were sometimes encapsulated by connective tissue. In the test group, the PLAB membrane remained basically unaltered in most of the samples during both healing periods, while in the positive test group, although the membrane was recognizable at 4 weeks of healing, it was partially resorbed or even undetectable in the late healing period at 12 weeks.

In the negative control group (the PLAB membrane alone), newly formed bone was observed in contact with the implant surface, and the proportion of new mineralized tissue was higher when compared with the tested groups using DBBM, although its thickness was considerably inferior. In all groups, a space between the bone and the membrane was frequently observed, which may correspond to either immature bone not detectable in the histological preparations or a dense periosteum-like tissue.

After 4 weeks of healing, new bone in contact with the implant surface at the dehiscence area could be identified in both the test and positive control group. In both groups, the presence of bone substitute particles occupied most of the space under the membrane, with them being surrounded by bone. In the negative control group, the apical area of the dehiscence was filled with new bone; in the coronal area, the membrane had collapsed into the implant surface, with the space between the membrane and implant occupied by soft tissue at this level.

In the PLAB membrane groups, the membrane structure and space were preserved, while in the sections with a native collagen membrane, its presence was frequently disintegrated, exhibiting tissue interstices between the membrane segments. In all groups, periosteum-like tissue was observed underneath the barrier membrane (Figure 6).

A net reduction in thickness was noted in all groups between 4 and 12 weeks. Hard tissue modeling was observed in most of the specimens using a bone replacement graft, especially bone apposition around the xenograft particles. In the negative control group, there was scarce additional bone modeling. At 12 weeks, the collagen membrane appeared in most of the sections as completely resorbed, while the PLAB membrane was still present, although with a reduced thickness (Figure 7). In most of the sections, there was an apical displacement of the bone replacement graft.

### 3.2. Histo-Morphometric Measurements

#### 3.2.1. Newly Formed Bone at 4 Weeks

The width of the newly formed buccal bone width at 1–2 and 3 mm from the implant platform is presented in Table 1 for the three tested groups. Although the differences were not statistically significant, the positive control group (NPCM) attained a higher width of new bone compared with the test group (PLAB) and the membrane-only group.

#### 3.2.2. Newly Formed Tissue Thickness at 4 Weeks

Differently from the results for the newly formed buccal bone width, during this healing period, the PLAB membrane group attained wider tissue thickness compared with the positive and negative control groups (Figure 8). At 3 mm from the implant platform, higher values were obtained by the test group, followed by the positive control and the negative control, (2.52 ± 0.9) vs. (2.50 ± 0.54) and (1.04 ± 0.63), with the difference between the test and negative control group being statistically significant (*p* = 0.03) (Table 2).

#### 3.2.3. Newly Formed Buccal Bone at 12 Weeks

During this healing period, the differences among the groups regarding the width of newly formed bone were reduced (Figure 9). An apical displacement of the bone replacement graft was clearly noticeable in both the test and positive control group. Table 2 depicts the linear measurements of increased width at the three height levels, with consistently higher widths in the positive control group, although very similar to the test group, followed by the negative control group. These differences were not statistically significant.

#### 3.2.4. Newly Formed Tissue Thickness at 12 Weeks

Compared to the 4-week measurements, clear tissue shrinkage had occurred in the three groups at all measured levels (Figure 10). The attained increased tissue thickness was similar between the test and positive control group (Table 2).

### 3.3. Bone to Implant Contact

At 4 weeks of healing, the highest BIC was obtained by the test group (39.54 ± 48.7) and the negative control (31.24 ± 42.6), while the more discrete results were obtained by the positive control (20.23 ± 36.1).

After 12 weeks of healing, the positive control group experienced an improvement in BIC, obtaining the best results (35.91 ± 24.9) while test and negative control groups experienced a decrease in BIC, (18.41 ± 20.5) and (24.3 ± 32.1) respectively.

### 3.4. Immunohistochemical Analysis

ALP was detected mainly around newly formed bone and bone substitutes. In all the groups the stain was more intense after 4 weeks of healing, while it decreased after 12 weeks of healing. The positive control group showed higher ALP reactivity than the test and negative controls at 4 weeks, although the differences were not statistically significant (5.25 ± 4.09, 4.46 ± 3.03, and 4.35 ± 2.28%, respectively, *p* > 0.05). After 12 weeks of healing the highest results were obtained in the positive control group followed by the test group and positive control (4.3 ± 2.14, 3.21 ± 1.53, and 2.39 ± 1.03%, respectively, *p* > 0.05).

## 4. Discussion

The present pre-clinical investigation evaluated the performance of three different guided bone regenerative procedures aimed at lateral bone augmentation using an experimental in vivo model of peri-implant dehiscence. The use of the DBBM + PLAB membrane showed a trend towards higher performance in terms of space maintenance capacity at the early healing stage, while, after 12 weeks of healing, these differences were lost with even a slightly better result in terms of newly formed bone and space maintenance with the collagen membrane + DBBM, although these differences were not statistically significant.

The available scientific evidence reports that native collagen membranes have a fast resorption rate, albeit maintaining their structural integrity between 4–8 weeks [31,32]. These results have been corroborated in the present investigation where the collagen membrane structure could be identified at 4 weeks but was fully or partially disintegrated at 12 weeks. Another potential disadvantage of native collagen membranes is their low stiffness and, hence, their lack of maintenance-keeping ability [10,33,34]. This fact was also corroborated in this investigation since the tissue thickness attained in this group at 4 weeks was clearly inferior to the tested synthetic membrane.

The synthetic polymer membrane tested (PLAB) has been reported to start its degradation after 6 weeks, with membrane remnants still being identifiable even after 12 months of healing [19]. Preclinical studies comparing the resorption rate of PLA and collagen membranes have concluded that PLA membranes will maintain their integrity for longer [35,36]. In the present study, complete integrity of the PLAB membrane was observed in the histological sections after 4 weeks of healing, while the rest of the membrane was still present in the 12-week sections. These synthetic bio-absorbable membranes (PLAB) have reported superior space maintenance properties [20,21], which is congruent with the results from the present investigation at the early healing stage, PLAB membranes demonstrated better results in bone and augmented tissue thickness, while the integrity of the membrane was maintained (4 weeks). However, after 12 weeks of healing, the PLA membrane lost its integrity, probably due to its mechanism of biodegradation. While collagen membranes are bio-absorbed by enzymatic degradation [3], synthetic polymer membranes like PLAB, when inserted into an aqueous environment, are degraded by hydrolysis [20,23]. This hydrolysis process leads to reduced PH levels because of the release of acidic degradation products, which may induce an inflammatory response [37,38]. This possible inflammatory response could explain the reason why after 12 weeks of healing, the group of the DBBM + PLAB membrane experienced a remarkable decrease in bone width, compared with the DBBM + collagen membrane group.

However, in both tested groups, there was a reduction in the width of bone and tissue thickness between 4–12 weeks. This finding may be explained by the tissue shrinkage that usually occurs in the late healing phases when the treated defects are over-contoured during the bone regenerative intervention with the bone replacement graft outside the bony envelope. The fact that the test group demonstrated greater shrinkage compared with the positive control group using the same bone replacement graft and a similar degree of over-contouring may be explained by differences in the membrane resorption process [19,31,38].

This tissue shrinkage between T4 and T12 was more pronounced coronally, with displacement of the BRG granules apically. This result is also consistent with previous similar pre-clinical investigations [24] and can be explained by the prolonged action of the masticatory forces, with them being more intense in the coronal aspect, thus pushing the biomaterial to the apical and lateral directions [39,40].

When the PLAB membrane was used without a bone replacement graft (negative control group), there was marked new bone formation after 4 and 12 weeks of healing, which corroborates both the barrier membrane and enhanced space-maintaining properties of this synthetic membrane. In fact, in some sections, the amount of new bone formation was superior in the membrane-only group, compared with the groups using a BRG. Similar results have been reported in other pre-clinical investigations where more pronounced bone formation with a membrane alone occurred mainly in distal defects [41], thus implying that a more favorable defect morphology with thicker buccal bone plates may enhance GBR with only a BM since the space maintenance is improved. Also, similar heterogeneous results have been reported in clinical studies using a BM with enhanced space-keeping properties alone in GBR simultaneous with implant placement. This study reported enhanced bone regeneration in some cases, while in others, the space under the membrane was fully collapsed onto the implant surface, thus recommending the use of a BRG for more consistent results [42].

Although in this investigation we did not test the behavior of the use of collagen membranes without a bone replacement graft, a previous report from our research group using a similar experimental model clearly demonstrated their barrier membrane effect, by demonstrating new bone formation under the membrane [9]. Conversely, results from a similar pre-clinical in vivo experiment, where no membrane and no bone replacement graft were used, the dehiscence defects remained unrepaired in the negative control group, with minimal natural bone healing occurring at the base of the defect [9]. The results from these reports, together with those reported in the present study, strengthen the importance of membrane barrier function to allow new bone formation by preventing the ingrowth of connective tissue into the defect.

In the immunohistochemical analysis, we used ALP as a recognized biomarker of osteoblast metabolism [43]. Previous investigations have reported that ALP expression after implant placement reaches its peak after 5–20 days [44], which correlates with the primary mineralization around dental implants. In a similar study performed by our research group [28], higher values of ALP expression were shown after 8 weeks of healing, and the ALP staining was still present after 16 weeks of healing. These results have been corroborated in the present investigation, where ALP activity was still present at 12 weeks, albeit reduced compared with the 4-week healing time.

The recent formulation of non-cross-linked collagen membranes demonstrates notable improvements in degradation time, mechanical properties, and an extended degradation period which enhanced the mechanical characteristics, including higher tensile strength and suture pullout strength [45]. Nevertheless, no differences were determined when compared with the standard collagen membranes in dehiscence-type models [46].

In a similar study to the present one, Al-Hazmi [46] evaluated different regenerative strategies in dehiscence-type defects in dogs by means of micro-CT analysis, obtaining the best results when a collagen membrane was used over the biomaterial. In another study, evaluating dehiscence-type defects in dogs, Jung 2017 obtained higher histometric results when a cross-linked collagen membrane was used (1.22 ± 0.53 mm) than in the positive and negative controls (0.42 ± 0.51 and 0.36 ± 0.50 mm, respectively) [9].

Although this in vivo preclinical investigation has clearly corroborated previous reports on the importance of using a BM in GBR simultaneous with implant placement, its results should be taken with caution in part due to the inherent limitations of the experimental model used, mainly in relation to its translation to humans. Nevertheless, it is important to highlight that the bone defects were left to chronified, thus mimicking a two-wall bone defect, which is frequently found in clinical human conditions. Another limitation in this experimental model is due to the possible influence of the bone defect position (mesial, medial, or distal), since distal defects are usually surrounded by thicker buccal bone plates, while mesial defects are contained in thinner bonny envelopes. It has been demonstrated that the thickness of the base of the defect may influence the level of bone regeneration [42]. Although randomization also accounted for the defect position, this is an intrasubject condition that may influence the results in part.

In spite of these limitations, we can conclude that new bone formation can be achieved in GBR procedures simultaneous with implant placement either with the use of a PLAB membrane or a native collagen membrane. Although in the early healing stages, the PLAB membrane seemed to attain higher bone and tissue width when compared to native collagen membranes, the increased shrinkage that occurred in the PLAB group reversed the results in the late healing period, although these differences were not statistically significant.

## Figures and Tables

**Figure 1 bioengineering-10-00841-f001:**
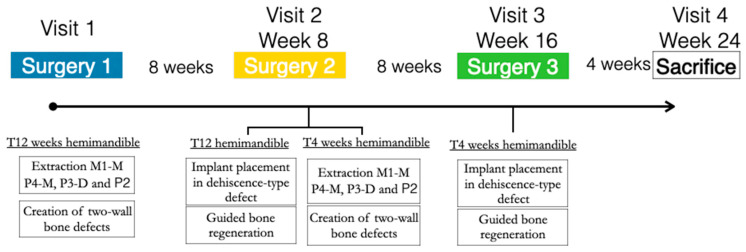
Study flowchart.

**Figure 2 bioengineering-10-00841-f002:**
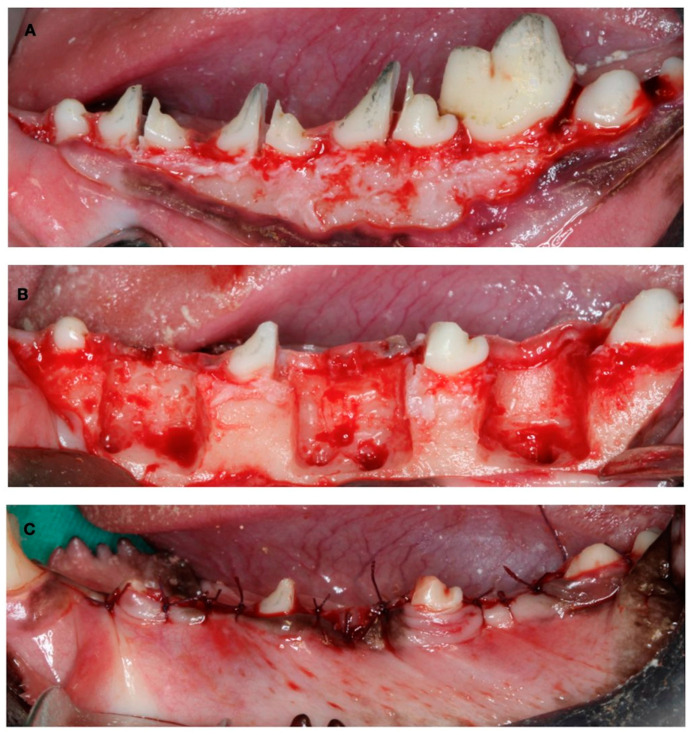
Surgical defect creation. (**A**) Tooth hemisection. (**B**) Tooth extraction and two-wall bone defect creation. (**C**) Suture.

**Figure 3 bioengineering-10-00841-f003:**
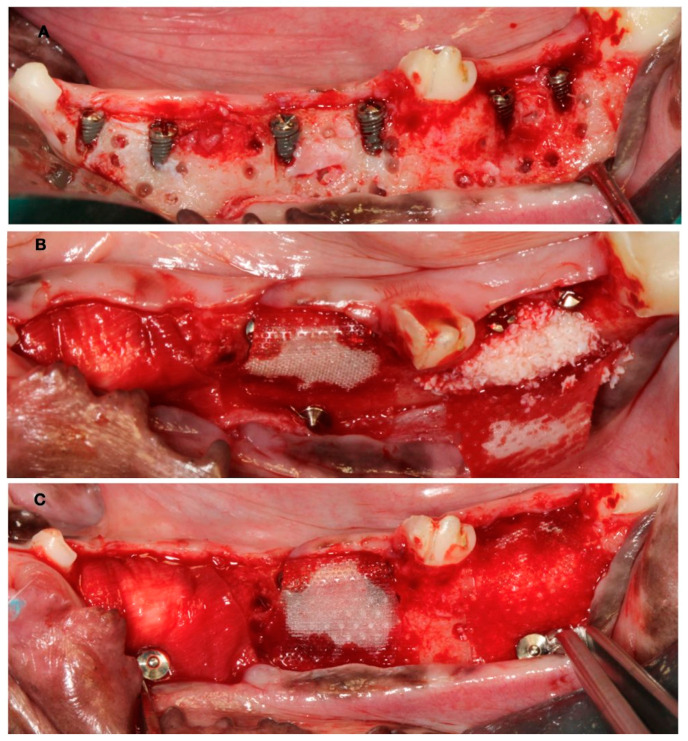
(**A**) Implant placement with vestibular dehiscence and bone perforations. (**B**) Detail of the three regeneration approaches. (**C**) Membrane stabilization with pins.

**Figure 4 bioengineering-10-00841-f004:**
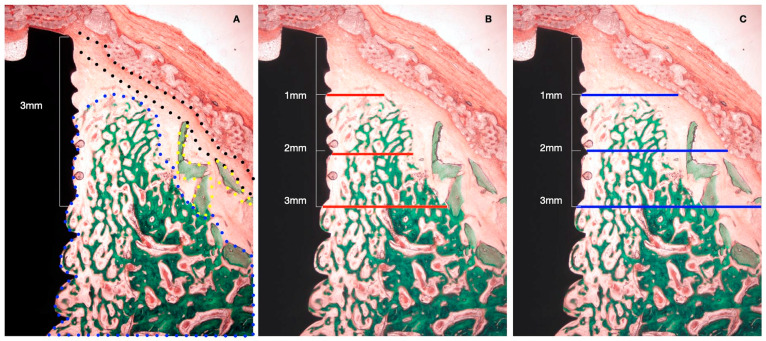
Histometric measurements of bone width and membrane width at 1, 2, and 3 mm. (**A**) Histological tissue identification (NB: newly formed bone, blue dots. XG: xenograft particles, yellow dots. MB: barrier membrane, black dots). (**B**) Histometrical measurement: newly formed bone. (**C**) Histometrical measurement: barrier space maintenance (augmented tissue thickness).

**Figure 5 bioengineering-10-00841-f005:**
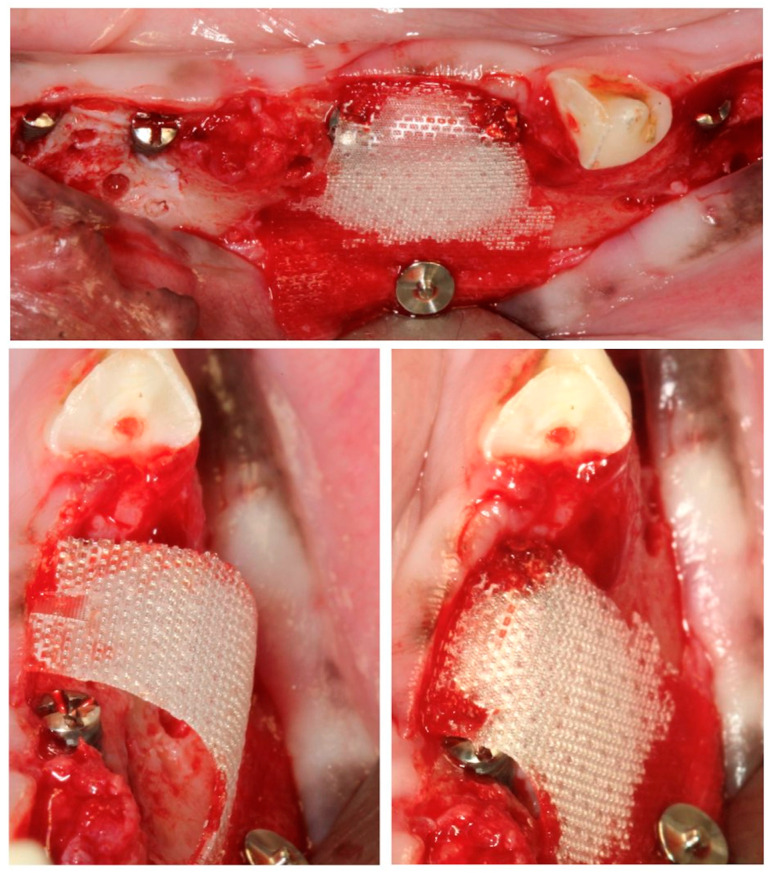
Space maintenance properties of the PLAB membrane when used without the mechanical support of a biomaterial underneath.

**Figure 6 bioengineering-10-00841-f006:**
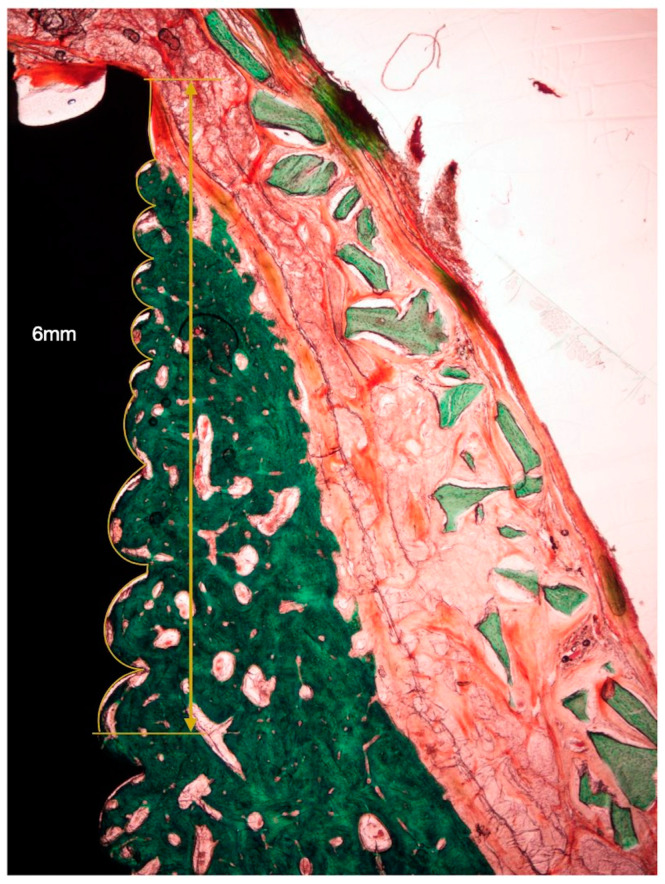
Histological bone-to-implant contact measurement. BIC: bone in contact with the implant surface within 6 coronal millimeters from the implant shoulder.

**Figure 7 bioengineering-10-00841-f007:**
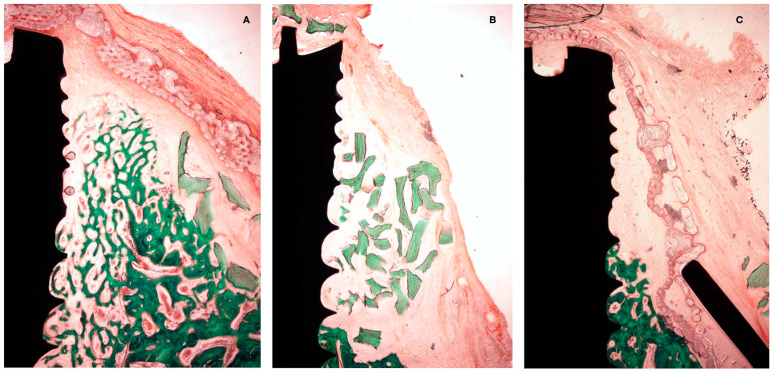
Histological preparation at 4 weeks of healing. (**A**) Test group. (**B**) +C group. (**C**) -C group.

**Figure 8 bioengineering-10-00841-f008:**
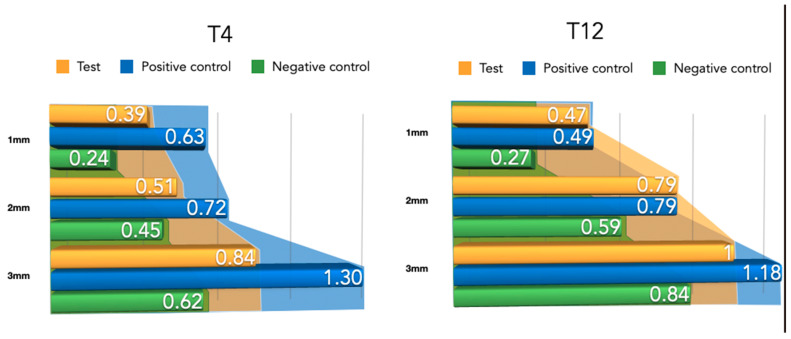
Newly formed bone thickness after 4 and 12 weeks of healing.

**Figure 9 bioengineering-10-00841-f009:**
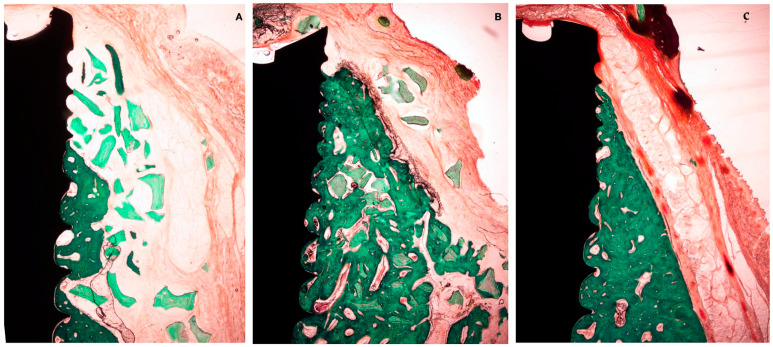
Histological preparation at 12 weeks of healing. (**A**) Test group. (**B**) +C group. (**C**) -C group.

**Figure 10 bioengineering-10-00841-f010:**
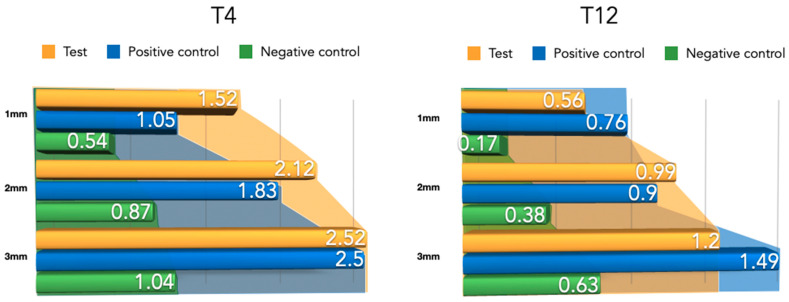
Barrier space maintenance (augmented tissue thickness) after 4 and 12 weeks of healing.

**Table 1 bioengineering-10-00841-t001:** Newly formed bone thickness measurements for both healing periods (4–12 weeks) at 1 mm, 2 mm, and 3 mm apico-coronal levels. All measurements are expressed in millimeters (mean ± SD values).

Group	Healing Time	1 mm	2 mm	3 mm
Test Group	4 weeks	0.39 ± 0.68	0.79 ± 1.01	0.84 ± 0.93
+C Group	4 weeks	0.63 ± 0.14	0.72 ± 0.16	1.30 ± 0.26
-C Group	4 weeks	0.24 ± 0.21	0.45 ± 0.41	0.62 ± 0.62
Test Group	12 weeks	0.47 ± 0.62	0.79 ± 1.01	100 ± 1.19
+C Group	12 weeks	0.49 ± 0.64	0.79 ± 0.82	1.18 ± 0.79
-C Group	12 weeks	0.27 ± 0.25	0.59 ± 0.75	0.84 ± 0.60

* Statistically significant differences between the groups at 4 weeks (*p* < 0.05). (No statistically significant differences were found). † Statistically significant differences between the groups at 12 weeks (*p* < 0.05) (No statistically significant differences were found).

**Table 2 bioengineering-10-00841-t002:** Augmented tissue thickness (barrier space maintenance) measurements for both healing periods (4–12 weeks) at 1 mm, 2 mm, and 3 mm apico-coronal levels. All measurements are expressed in millimeters (mean ± SD values).

Group	Healing Time	1 mm	2 mm	3 mm
Test Group	4 weeks	1.52 ± 1.12	2.20 ± 0.89	2.52 ± 0.90
+C Group	4 weeks	1.05 ± 0.82	1.83 ± 0.77	2.50 ± 0.54
-C Group	4 weeks	0.54 ± 0.61	0.87 ± 0.64	1.04 ± 0.63
Test Group	12 weeks	0.56 ± 0.64	0.99 ± 1.02	1.20 ± 1.08
+C Group	12 weeks	0.76 ± 0.97	0.90 ± 1.02	1.49 ± 0.96
-C Group	12 weeks	0.17 ± 0.27	0.38 ± 0.42	0.63 ± 0.62

* Statistically significant differences between the groups at 4 weeks (*p* < 0.05). † Statistically significant differences between the groups at 12 weeks (*p* < 0.05) (no statistically significant differences were found).

## Data Availability

Not applicable.

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
