# Peer review of "A Synthetic Bio-Absorbable Membrane in Guided Bone Regeneration in Dehiscence-Type Defects: An Experimental In Vivo Investigation in Dogs"

_bioengineering, 2023, doi:10.3390/bioengineering10070841_

Round 1

Reviewer 1 Report

Authors have attempted to determine the performance of a polylactic acid and acetyl-butyl-citrate synthetic barrier membrane for lateral bone augmentation of peri-implant dehiscence defects. The experiment has been well-designed, executed and well-presented.

Few areas which could be improved for better clarity and understanding as per my observations are as follows:

1. It is interesting to note that a similar study by the same research group has been published (https://doi.org/10.1007/s00784-020-03537-5). It appears that it is the same experiment, bearing the same Animal Ethical Committee approval number, where the Micro CT and soft tissue parameters have been reported.

Although the authors have cited this work at two places in the manuscript, there is a lack of clarity that a part of the same experiment has been already presented. It is suggested that the authors include this information in the introduction section and present the current data as histological and histomorphometric findings of the same experiment. It is also suggested that the title be suitably modified.

Line 376-377; Authors are comparing the findings of the same experiment. It is suggested that authors discuss by providing a valid and relevant reference.

Figure 3: Shows A, B and C in the composite image.  Authors should provide the details of the image in the description

2. Figure 1: Authors mention creation of horizontal defects. The termininology horizontal defects does not seem to be appropriate in the context of the experiment. Authors have also used a term 'defect boxes', ' peri-implant defects' in this manuscript and the related work. It is suggested that an appropriate and consistent terminology be used.

3. Fig 1 uses several terminologies at the same time such as Week, Visits and Surgery, however, terminologies T4 and T12 have neither been mentioned in Fig 1 nor explained elsewhere  in the manuscript. It is suggested that these terms be explained, indicated on the timeline and detailed in figure 8 and 10.

4. Figure legends of Figure 5 - Figure 10 can be improved for clarity

5. The measurement values depicted on schematics on Fig 8 and Fig 10 needs to be in a contrast color. It is not visible.

6. It is suggested to follow one scheme of writing the decimals, either 0.65 or 0,65. 

Minor Grammatical corrections needed

Author Response

Dear reviewer, thank you for your comments and remarks. In the following lines I will try to answer them. All the changes are highlighted in the text.

1.

A) We have clarified that both manuscripts belong to the same study and added the following sentence to the lines 86-88. “The present work is a complementary analysis to the previously published Micro CT and profilometric data analysis (Di Raimondo et al., 2020). The present report provides the histologic results of the same experimental study”.

B) Dear reviewer, thank you for your query, we have added the corresponding references (19,31,38) and highlighted on the manuscript.

C)The following details have been added to the image caption:

    A: Implant placement with vestibular dehiscence and bone perforations.

    B: Detail of the three regenerations approaches

    C: Membrane stabilization with pins

2.  Terms have been changed into dehisence-type defects or two-wall bone defects respectively to make it clearer as reviewer suggested

3. Images and captions were changed accordingly. (Fig 1)

Figure 8. Newly formed bone thickness after 4 and 12 weeks healing.

Figure 10. Barrier space maintenance (Augmented tissue thickness) after 4 and                     12 weeks healing.

4. Legends were changed to improve clarity

Figure 5. Space maintenance properties of PLAB membrane when used without the mechanical support of biomaterial underneath.

Figure 10. Barrier space maintenance (Augmented tissue thickness) after 4 and 12 weeks healing.

5.  Figures were changed in order to make numbers more visible.

6. Decimals were all congruently changed into dots and not comas.

I would like to thank you once again for all the convenient comments. 

Best regards

Reviewer 2 Report

The authors presented a very interesting in vivo study about GBR, reporting histological and histomorphometric results. The quality of the paper and the clarity of writing as well as the organization of the paper was very good. I recognize also the appropriateness of approach / study design and the overall importance of the work for the scientific community. However, the authors exclude to discuss their results respect to those reported in similar study using different biomaterial, i.e. cross-linked collagen membrane or synthetic graft. A complete absence of references to the micro-CT analysis is an important limitation of this study. In order to avoid an author bias, I suggest to improve the paper with the following issues.

TITLE

I suggest to add “dehiscence-type defects” and “in dogs”.

ABSTRACT

Please report the standardized extension of implant dehiscences in millimeters.

The presentation of results seem lead to an authors bias. Since no statistical differences were observed, please report the mean valus and S.D. and P-values.

The use of “best results” is misunderstanding and confounding for the readers. Remove them from abstract.

INTRODUCTION

Please summarize the first paragraphs ( “The rehabilitation of the lost dentition… 2.73 mm (95% CI: 2.36–3.11), 1.71 mm (95% CI: 1.30–46 2.12) and 1.44 mm (95% CI: 0.78–2.10), respectively” ).

Alveolar remodelling after extraction is not the issue of the manuscript.

Also this paragraphs reported basic and well-known information about GBR and they could be summarized (“The bone replacement graft serves as …. to repopulate the defect and generate new bone.”)

A dissertation about cross-linked collagen membranes could be add in introduction as well as in the discussion. The authors could cite these paper about cross-linked membranes.

-       Schwarz F, Sahm N, Becker J. Impact of the outcome of guided bone regeneration in dehiscence-type defects on the long-term stability of peri-implant health: clinical observations at 4 years. Clin Oral Implants Res. 2012 Feb;23(2):191-196.

-       Omar O, Dahlin A, Gasser A, Dahlin C. Tissue dynamics and regenerative outcome in two resorbable non-cross-linked collagen membranes for guided bone regeneration: A preclinical molecular and histological study in vivo. Clin Oral Implants Res. 2018 Jan;29(1):7-19.

-       Cucchi A, Sartori M, Aldini NN, Vignudelli E, Corinaldesi G. A Proposal of Pseudo-periosteum Classification After GBR by Means of Titanium-Reinforced d-PTFE Membranes or Titanium Meshes Plus Cross-Linked Collagen Membranes. Int J Periodontics Restorative Dent. 2019 Jul/Aug;39(4):e157-e165.

-       Lee SR, Jang TS, Seo CS, Choi IO, Lee WP. Hard Tissue Volume Stability Effect beyond the Bony Envelope of a Three-Dimensional Preformed Titanium Mesh with Two Different Collagen Barrier Membranes on Peri-Implant Dehiscence Defects in the Anterior Maxilla: A Randomized Clinical Trial. Materials (Basel). 2021 Sep 27;14(19):5618.

The aims of the study should include the linear measurements too. Therefore, more details about the investigated outcomes could be described in the final part of introduction: primary outcomes and secondary outcomes.

MATERIALS AND METHODS

Did the authors evaluate the features of dehiscence before GBR?  

STATISTICAL ANALYSIS AND RESULTS

I agree with statistical methods, but I suggest to report the confidential intervals (C.I. 95%) that can better express the variability of the outcomes.

DISCUSSION

Histological results should be discussed on the basis of implant surface, as influencing factor for the healing of bone around implants.

For examples, Schwarz F, Herten M, Sager M, Wieland M, Dard M, Becker J. Bone regeneration in dehiscence-type defects at chemically modified (SLActive) and conventional SLA titanium implants: a pilot study in dogs. J Clin Periodontol. 2007 Jan;34(1):78-86. doi: 10.1111/j.1600-051X.2006.01008.x. Epub 2006 Nov 24. PMID: 17137467.

Clinical results in the treatment of similar defects using novel native collagen membrane should be addressed.

For example, Urban IA, Wessing B, Alández N, Meloni S, González-Martin O, Polizzi G, Sanz-Sanchez I, Montero E, Zechner W. A multicenter randomized controlled trial using a novel collagen membrane for guided bone regeneration at dehisced single implant sites: Outcome at prosthetic delivery and at 1-year follow-up. Clin Oral Implants Res. 2019 Jun;30(6):487-497.

The discussion should include the issue of cross-linked collagen membranes, that have features and properties different from those of synthetic or native membranes. See comment above.

Nowadays, the histological and histomorphometrical analysis as well as bone implant contact could be evaluated 3D using micro-CT analysis in vivo as well as in humans. This analysis allows to have 3D data, a comprehensive evaluation of the samples (VOI vs. ROI), and an automatic measurements of morphometric parameters. Please discuss this issue and reported this as a limitation of this study and a suggestion for further research.

For example, the authors can discuss and cite some of the following in vivo studies based on micro-CT analysis:

-       In dogs - Al-Hazmi BA, Al-Hamdan KS, Al-Rasheed A, Babay N, Wang HL, Al-Hezaimi K. Efficacy of using PDGF and xenograft with or without collagen membrane for bone regeneration around immediate implants with induced dehiscence-type defects: a microcomputed tomographic study in dogs. J Periodontol. 2013 Mar;84(3):371-8.

-       In minipigs - Dahlin C, Obrecht M, Dard M, Donos N. Bone tissue modelling and remodelling following guided bone regeneration in combination with biphasic calcium phosphate materials presenting different microporosity. Clin Oral Implants Res. 2015 Jul;26(7):814-22.

-       In rats - Chou J, Komuro M, Hao J, Kuroda S, Hattori Y, Ben-Nissan B, Milthorpe B, Otsuka M. Bioresorbable zinc hydroxyapatite guided bone regeneration membrane for bone regeneration. Clin Oral Implants Res. 2016 Mar;27(3):354-60.

-       In rats - Basudan A, Babay N, Ramalingam S, Nooh N, Al-Kindi M, Al-Rasheed A, Al-Hezaimi K. Efficacy of Mucograft vs Conventional Resorbable Collagen Membranes in Guided Bone Regeneration Around Standardized Calvarial Defects in Rats: An In Vivo Microcomputed Tomographic Analysis. Int J Periodontics Restorative Dent. 2016;36 Suppl:s109-21.

-       In dogs - Hsu YT, Al-Hezaimi K, Galindo-Moreno P, O'Valle F, Al-Rasheed A, Wang HL. Effects of Recombinant Human Bone Morphogenetic Protein-2 on Vertical Bone Augmentation in a Canine Model. J Periodontol. 2017 Sep;88(9):896-905.

-       In rabbits - Kobayashi E, Fujioka-Kobayashi M, Saulacic N, Schaller B, Sculean A, Miron RJ. Effect of enamel matrix derivative liquid in combination with a natural bone mineral on new bone formation in a rabbit GBR model. Clin Oral Implants Res. 2019 Jun;30(6):542-549.

-       In rabbits - Jung J, Park JS, Dard M, Al-Nawas B, Kwon YD. Effect of enamel matrix derivative liquid combined with synthetic bone substitute on bone regeneration in a rabbit calvarial model. Clin Oral Investig. 2021 Feb;25(2):547-554.

-       In dogs - Thieu MKL, Haugen HJ, Sanz-Esporrin J, Sanz M, Lyngstadaas SP, Verket A. Guided bone regeneration of chronic non-contained bone defects using a volume stable porous block TiO2 scaffold: An experimental in vivo study. Clin Oral Implants Res. 2021 Mar;32(3):369-381.

-       In rabbits - Yun J, Lee J, Ha CW, Park SJ, Kim S, Koo KT, Seol YJ, Lee YM. The effect of 3-D printed polylactic acid scaffold with and without hyaluronic acid on bone regeneration. J Periodontol. 2022 Jul;93(7):1072-1082.

For example, the authors can discuss and cite some of the following clinical studies based on micro-CT analysis:

-       Maxillary sinus lift - Soardi CM, Clozza E, Turco G, Biasotto M, Engebretson SP, Wang HL, Zaffe D. Microradiography and microcomputed tomography comparative analysis in human bone cores harvested after maxillary sinus augmentation: a pilot study. Clin Oral Implants Res. 2014 Oct;25(10):1161-8.

-       Maxillary sinus lift: Huang HL, Hsu JT, Chen MY, Liu C, Chang CH, Li YF, Chen KT. Microcomputed tomography analysis of particular autogenous bone graft in sinus augmentation at 5 months: differences on bone mineral density and 3D trabecular structure. Clin Oral Investig. 2013 Mar;17(2):535-42.

-       Socket preservation: Villanueva-Alcojol L, Monje F, González-García R, Moreno C, Monje A. Characteristics of newly formed bone in sockets augmented with cancellous porous bovine bone and a resorbable membrane: microcomputed tomography, histologic, and resonance frequence analysis. Implant Dent. 2013 Aug;22(4):380-7.

-       Socket preservation: Neiva R, Pagni G, Duarte F, Park CH, Yi E, Holman LA, Giannobile WV. Analysis of tissue neogenesis in extraction sockets treated with guided bone regeneration: clinical, histologic, and micro-CT results. Int J Periodontics Restorative Dent. 2011 Sep-Oct;31(5):457-69.

-       GBR - Cucchi A, Vignudelli E, Sartori M, Parrilli A, Aldini NN, Corinaldesi G. A microcomputed tomography analysis of bone tissue after vertical ridge augmentation with non-resorbable membranes versus resorbable membranes and titanium mesh in humans. Int J Oral Implantol. 2021 Mar 16;14(1):25-38.

-       Onlay - Wang J, Luo Y, Qu Y, Man Y. Horizontal ridge augmentation in the anterior maxilla with in situ onlay bone grafting: a retrospective cohort study. Clin Oral Investig. 2022 Sep;26(9):5893-5908.

I suggest to improve the discussion comparing their results to those reported by other in vivo studies about GBR using different biomaterial or different membranes.

OTHERS

Conflict of interest and financial support should be addressed.

Author Response

Dear reviewer, 

Thank you for all the convenient comments as well as for all the interesting and useful references. 

I attach a file with all the answer and another with the manuscript with the changes. 

Round 2

Reviewer 1 Report

Satisfactory Changes Made.

Suggested Title: "Synthetic Bioabsorbable Membrane in Guided bone regeneration in dehiscence-type defects. A histological and histomorphometric analysis of experimental in-vivo investigation in dogs."